# Skin coloration is a culturally-specific cue for attractiveness, healthiness, and youthfulness in observers of Chinese and western European descent

Yan Lu[1], Jie Yang[1,2], Kaida Xiao[1], Michael Pointer[1], Changjun Li[3], Sophie Wuerger[4]*

**1** School of Design, University of Leeds, Leeds, United Kingdom, **2** School of New Media, Beijing Institute of Graphic Communication, Beijing, China, **3** School of Computer Science and Software Engineering, University of Science and Technology Liaoning, Anshan, China, **4** Department of Psychology, University of Liverpool, Liverpool, United Kingdom

* s.m.wuerger@liverpool.ac.uk

**Data Availability Statement:** The data associated with this research will be available at https://pcwww.liv.ac.uk/~sophiew/skin.htm.

## Abstract

Facial skin coloration signals information about an individual and plays an important role in social interactions and mate choice, due its putative association with health, attractiveness, and age. Whether skin coloration as an evolutionary significant cue is universal or specific to a particular culture is unclear and current evidence on the universality of skin color as a cue to health and attractiveness are equivocal. The current study used 80 calibrated, high-resolution, non-manipulated images of real human faces, either of Chinese or western European descent, which were rated in terms of attractiveness, healthiness, and perceived age by 44 observers, 22 western European (13 male; mean age ± SD = 24.27 ± 5.30) and 22 Chinese (7 male; mean age ± SD = 26.05 ± 3.96) observers. To elucidate the associations between skin coloration and these perceptual ratings and whether these associations are modulated by observer or image ethnicity, a linear mixed-effect model was setup with skin lightness (L*; CIELAB), redness (a*) and yellowness (b*), observer and image ethnicity as independent variables and perceived attractiveness, healthiness, and estimated age as dependent variables. We found robust positive associations between facial skin lightness (L*) and attractiveness, healthiness, and youthfulness, but only when Chinese observers judge facial images of their own ethnicity. Observers of European descent, on the other hand, associated an increase in yellowness(b*) with greater attractiveness and healthiness in Chinese facial images. We find no evidence that facial redness is positively associated with these attributes; instead, an increase in redness (a*) is associated with an increase in the estimated age of European facial images. We conclude that observers of both ethnicities make use of skin color and lightness to rate attractiveness, healthiness, and perceived age, but to a lesser degree than previously thought. Furthermore, these coloration cues are not universal and are utilized differently within the Chinese and western European ethnic groups. Our study adds to the growing body of work demonstrating the importance of skin color manipulations within an evolutionary meaningful parameter space, ideally using realistic skin models based on physical parameters.

**Funding:** The database used in this experiment was generated with funding from the Engineering and Physical Sciences Research Council (EPSRC) [grant number EP/K040057 awarded to SW]. www. epsrc.ac.uk The funder had no role in the study design, data collection and analysis, decision to publish, or preparation of the manuscript.

**Competing interests:** The authors have declared that no competing interests exist.

## Introduction

As one of the most significant features of a human face, skin coloration has been implied as a factor in sexual selection [1] and increased facial skin lightness, redness and yellowness, have been positively associated with healthy appearance and facial attractiveness [2, 3]. Facial redness in particular, has been shown to enhance perceived healthiness and attractiveness equally [4], possibly reflecting cardiovascular fitness of humans. Skin color homogeneity, driven by the distribution of skin chromophores melanin and hemoglobin, is positively associated with a younger perceived age and greater health and attractiveness [5–7].

In virtually all experiments, with the exception of several more recent studies [8–11], observers were either presented with color-manipulated, often morphed, facial images [4, 12, 13] or were asked to manipulate the facial image along fixed dimensions in color space, such that perceived healthiness or attractiveness was optimized [2, 3]. Computer-generated or morphed facial images, however, may lose skin texture and appear to be unrealistic after image processing. Furthermore, a uniform color shift applied to each pixel of the facial image, is not necessarily consistent with naturally occurring coloration changes since the variation in the color pattern depends on the distribution of blood vessels across the face which is not uniform. Human observers have been shown to be sensitive to these spatial color variations and perceived health is affected differentially by the color changes in particular facial regions [14].

With some notable exceptions, the impact of facial color on subjective attributes (perceived healthiness, attractiveness, and youthfulness) has been studied using samples of European descent, both as participants and as stimulus material. Ethnicity-specific effects have primarily focused on structural facial features; average features, neotenous features, and feminine features have been shown to affect perceived attractiveness [15–17]. When viewing black and white facial images, observers agreed more strongly in their attractiveness ratings for own-ethnicity faces [16, 17]. In relation to skin coloration as a cue to perceived health and attractiveness, similar preferences for skin coloration have been demonstrated in Caucasian and African samples and observers [18, 19], not only for their own but also across ethnicities. Mainland Chinese observers have been shown to prefer lighter skin and decreased yellowness in contrast to Caucasian observers, but only when judging faces of their own ethnicity [20]. In contrast, when asked to optimize perceived healthiness, Malaysian-Chinese observers increased facial yellowness and redness and decreased lightness, irrespective of the ethnicity of the faces which were Chinese-Malaysian, Caucasian or African [21]. One of the aims of the present study was to shed light on these mixed results and to dissociate the effect of observer and facial image ethnicity on these subjective judgements.

Some of the associations between skin color and perceptual ratings are clearly grounded in physical skin changes, e.g. the negative association between Lightness ($L^*$) and both physical and perceived age is linked to sun exposure over time [22]. Other associations between skin coloration and preference ratings might reflect the aesthetic differences between western and eastern culture, which are likely to result from the development of multiple social and cultural factors over a long period of time [23–26]. We therefore hypothesize that associations between skin coloration and preferences will depend on both, the ethnicity of the observer and the ethnicity of the viewed facial image.

The aim of our study was to establish (1) the strength of the association between skin coloration ($L^*$, $a^*$, $b^*$) and the perceptual attributes (attractiveness, healthiness, youthfulness), and (2) whether the utilization of these skin coloration cues ($L^*$, $a^*$, $b^*$) is modulated by the ethnicity of the observer, the viewed image, or both. We used a large set (n = 80) of calibrated, high-resolution images of real human faces (Chinese and western Europeans) comprising a range of ethnicity-specific skin colors (lightness $L^*$, redness $a^*$, yellowness $b^*$), without digitally manipulating skin color or texture. Two groups of observers (Chinese and western Europeans) rated

both sets of facial images along three dimensions: perceived attractiveness, healthiness, and youthfulness.

# Materials and methods

## Photography and image processing

All facial images used in this study were selected from the Liverpool-Leeds Skin-color Database (LLSD), which included data for 188 subjects from four ethnic groups (western European, Oriental, South Asian and African, including both genders) and was established by the Universities of Liverpool and Leeds [27, 28]. The facial image of each subject was obtained by photography in a VeriVide DigiEye® light booth, which provided a uniform matte mid-grey background and even, diffuse illumination that simulated CIE illuminant D65 (daylight). There was no other lighting in the room where the photography took place. During data collection, the participant sat in the viewing cabinet and their target facial area was adjusted to fit within the camera image. A digital SLR camera (Nikon D7000), controlled by the DigiEye system software, was used to capture images of training color charts for camera characterization and of each subject's face. The distance from the participant to the camera was approximately 57.5 cm and the participant looked straight into the camera.

Eighty real facial images, 40 Chinese images and 40 of western European descent, all with a neutral facial expression, were selected from the LLSD database for this study. All the faces are from people of the same age range between 20–40 and both genders were included in each ethnic group. The RGB data of each pixel of each image was first transformed to spectral reflectance [29] and later into CIELAB color coordinates [30]. To truly represent the color appearance of those facial images, a BenQ professional color display was used, with the white point set to D65 (the same as the illumination for facial image capturing). The method of piecewise linear interpolation assuming constant chromaticity (PLCC) [31, 32] was used for the color characterization of the display and the CIELAB values for each pixel were transformed to display RGB values for each facial image. Subsequently, each facial image was edited to remove the hair, ears, and any visible clothing manually and the image was then scaled to be in the center of the screen. Finally, a mid-grey background (L*, a*, b* = 50, 0, 0) was set to display all the images. All the images were processed in MATLAB. Fig 1 shows an example of a Chinese real facial image used in this study.

## Stimulus description

The mean color specification, in terms of CIELAB coordinates, of 80 test facial images (40 Chinese and 40 western Europeans) were calculated as the overall mean of each pixel in the facial area, excluding the mouth, nose, eyes, and eyebrows. CIELAB color space is a device-independent standard color appearance space where skin color is described by three dimensions: Lightness L*, Redness a* and Yellowness b* [33]. Additionally, chroma is used to roughly represent the saturation of colors. As shown in Fig 2, skin colors of both western Europeans and Chinese images were plotted in the a*-b* chromaticity diagram as well as the Chroma-Lightness diagram. There are systematic mean differences in lightness and chromaticity between the two ethnic groups. The western European images (average L* = 59.0, a* = 8.3, b* = 14.1) have, on average, higher lightness and lower yellowness (b*) compared to the Chinese images (average L* = 55.0, a* = 8.9, b* = 16.9).

## Observers

A psychophysical experiment was conducted using 44 observers, including 22 western Europeans (13 male; mean age ± SD = 24.27 ± 5.30) and 22 Chinese (7 male; mean

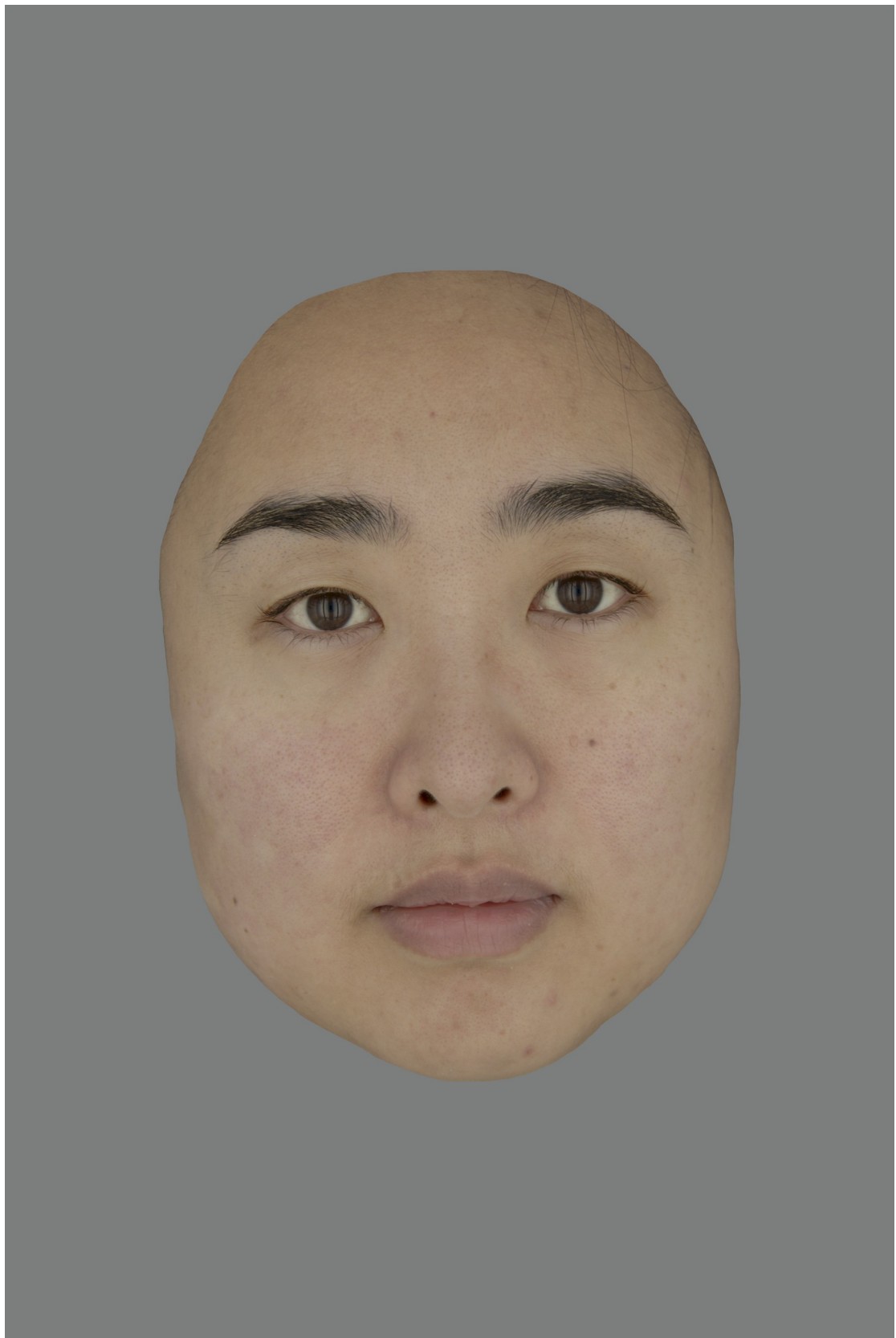

**Fig 1. An example of a Chinese facial image.**

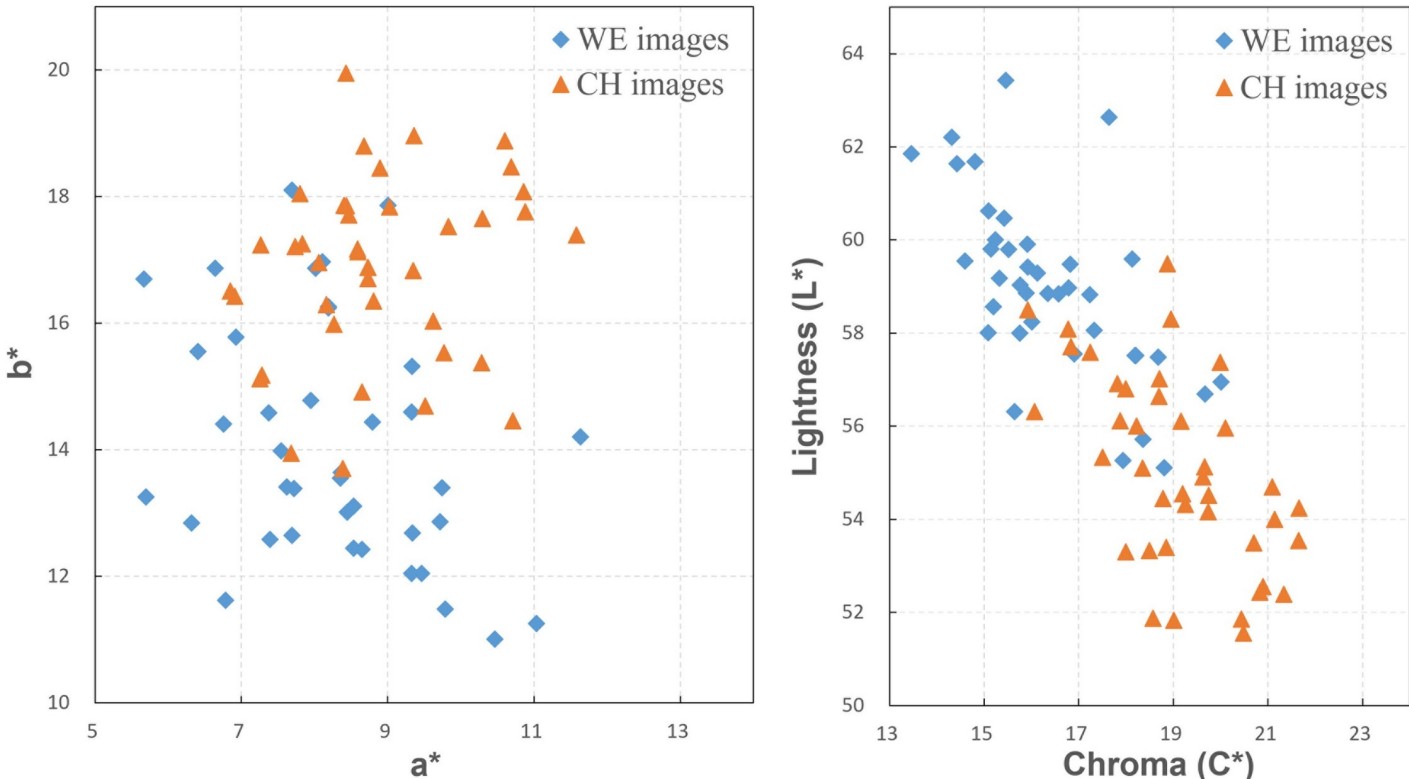

**Fig 2. The distribution of the mean facial colors of the test facial images in CIELAB a\*b\* space (left) and L\*C\* space (right).** ◆ Western European (WE), ▲ Chinese (CH).

age ± SD = 26.05 ± 3.96), who each evaluated the appearance of the 80 facial images (40 Chinese and 40 western Europeans) using three subjective attributes: perceived attractiveness, perceived healthiness, and perceived age. The sample size needed was estimated using G-power 3.1 software package [34] and a total sample of at least 34 was needed to ensure 80% power to detect a medium effect size of 0.5 at $p < 0.05$. All observers were given instructions in English and confirmed their understanding before the experiments. The Chinese observers were from mainland China, and at the time of the study, they were at Leeds University, UK as students or visiting scholars. On average, they spent about 1–3 years in UK. This study was approved by the Ethics Committee at the University of Leeds (LTDESN-090) and all participants gave written informed consent prior to taking part in the study. The individual in this manuscript has given written informed consent (as outlined in PLOS consent form) to publish these case details.

## Experimental procedure

The experiment was divided into three separate sessions. In each session, the observer viewed 80 facial images presented in random order and rated the skin color of each image with respect to one of the three attributes. The observer viewed each image for eight seconds and then made a judgement of the facial skin color without a time limit. The following question was asked after the observation of each image, "based on the skin color, what attractiveness score (or healthiness score or the estimated age, depend on different sessions) you would give for the last image?" Using a categorical judgment method, the perceived facial attractiveness and healthiness were rated on a 7-point Likert-type scale where 1 represented 'least attractiveness' /

'healthiness' and 7 represented 'best attractiveness' / 'healthiness'. The visual age was rated on a single-year step scale from 1 to 99 years. The sex effect was not tested in this study and both observer and image sex were balanced. The observers were not either told the image sex or asked to judge the image sex when making judgements. Since each facial image was edited to remove the hair, ears, and any visible clothing, the sex of the face was not that obvious or highlighted in this study.

## Statistical analysis

Inter-observer variability was examined by calculating Cronbach Alpha Coefficients [35]. For each of the three dependent variables (DV—attractiveness, healthiness, perceived age) a linear mixed-effect model was set up with the following fixed effects: lightness (L*), redness (a*) and yellowness (b*) as continuous predictors and image ethnicity and observer ethnicity as categorical predictors, including random intercepts for both images and observers.

All linear mixed-effect models were implemented in lme4 R package [36]. Deviation coding was used to convert both image ethnicity and observer ethnicity into deviation-coded factors (code '-0.5' for western European images/observers and code '0.5' for Chinese images/observers) for testing the main effects of each model. For each DV, a simple and a full model was considered; the full model allowing for all interactions between color (L*, a*, b*) and ethnicity (observer ethnicity, image ethnicity). In all cases the two model predictions differed significantly from each other and both AIC/BIC favored the model including the interactions [37]. P values for the fixed effects in each linear mixed-effect model were calculated using F tests with type III sums of squares and Satterthwaite's degrees-of-freedom approximation in the lmerTest R package [38]. Significant interactions revealed in the tests were followed up with a further analysis of the simple effects for each subgroup. In addition, Pearson's correlation coefficients (two-tailed) were also used to test the associations of the perceptions of all three attributes for both sets of observers.

## Results

### Consistency within and between ethnicities

The internal consistency in the ratings of attractiveness, healthiness, and age for both the western European and Chinese groups of observers is very high, with values of Cronbach's alpha coefficient all greater than 0.87 (Table 1). On average, both set of observers show high levels of agreement on the subjective ratings in their own-ethnicity faces and other- ethnicity faces.

**Table 1. The Cronbach Alpha Coefficient for assessing the inter-observer variability of the western European (WE) and Chinese (CH) observers (sample size).**

|  | WE | CH | WE & CH |
|---|---|---|---|
| **WE images** |  |  |  |
| Attractiveness | 0.96 (22) | 0.93 (22) | 0.96 (44) |
| Healthiness | 0.96 (22) | 0.93 (22) | 0.97 (44) |
| Age | 0.90 (22) | 0.91 (22) | 0.95 (44) |
| **CH images** |  |  |  |
| Attractiveness | 0.95 (22) | 0.96 (22) | 0.97 (44) |
| Healthiness | 0.96 (22) | 0.96 (22) | 0.98 (44) |
| Age | 0.87 (22) | 0.92 (22) | 0.94 (44) |

**Table 2. Model comparisons: Mixed models with and without interactions for all three attributes.**

| model | npar | AIC | BIC | logLik | deviance | $\chi^2$ | $\chi^2$df | P |
|---|---|---|---|---|---|---|---|---|
| **DV = attractiveness** | | | | | | | | |
| no interaction | 9 | 9725 | 9781 | -4854 | 9707 | | | |
| + interactions | 19 | 9534 | 9651 | -4748 | 9496 | 211.03 | 10 | <0.001*** |
| **DV = healthiness** | | | | | | | | |
| no interaction | 9 | 9471 | 9526 | -4726 | 9453 | | | |
| + interactions | 19 | 9344 | 9460 | -4653 | 9306 | 147.1 | 10 | <0.001*** |
| **DV = age** | | | | | | | | |
| no interaction | 9 | 20013 | 20068 | -9997 | 19995 | | | |
| + interactions | 19 | 20002 | 20119 | -9982 | 19964 | 31.09 | 10 | <0.001*** |

*P≤0.05

** P≤0.01

***P≤0.001.

## The effect of skin coloration, observer and image ethnicity on perceived healthiness, attractiveness, and age

For all perceptual attributes, we first evaluate a linear mixed-effect model with and without interactions. In all cases, a model allowing for interactions outperforms the model without interactions, as shown in Table 2. While the BIC for age weakly favors the model with no interaction, all the AIC strongly prefers the model with interactions, as does the significant likelihood ration test. We will therefore report the analysis of the model with interactions.

Table 3 shows all the main effects of the linear mixed-effect model for facial attractiveness. For attractiveness, neither skin coloration ($L^*$, $a^*$, $b^*$) nor observer ethnicity are significant, but

**Table 3. Linear mixed effects model estimates of fixed effects, their SE, t-value, lower (2.5%) and upper (97.5%) confidence intervals and P-values for attractiveness.**

| Fixed effects | Estimate | SE | *t*-value | 2.5% CI | 97.5% CI | P-value |
|---|---|---|---|---|---|---|
| (Intercept) | 2.346 | 4.465 | 0.525 | -6.513 | 11.206 | 0.601 |
| $L^*$ | 0.021 | 0.058 | 0.359 | -0.094 | 0.136 | 0.720 |
| $a^*$ | -0.113 | 0.085 | -1.328 | -0.282 | 0.056 | 0.188 |
| $b^*$ | 0.106 | 0.061 | 1.739 | -0.015 | 0.228 | 0.086 |
| Im | -20.703 | 8.929 | -2.319 | -38.419 | -2.987 | **0.023**\* |
| Ob | -1.531 | 1.780 | -0.860 | -5.020 | 1.958 | 0.390 |
| Im:Ob | -6.266 | 3.546 | -1.767 | -13.218 | 0.686 | 0.077 |
| $L^*$:Im | 0.284 | 0.116 | 2.448 | 0.054 | 0.515 | **0.017**\* |
| $L^*$:Ob | 0.077 | 0.023 | 3.353 | 0.032 | 0.123 | **0.001**\*\*\* |
| $a^*$:Im | 0.187 | 0.170 | 1.102 | -0.150 | 0.525 | 0.274 |
| $a^*$:Ob | -0.166 | 0.034 | -4.901 | -0.232 | -0.099 | **<0.001**\*\*\* |
| $b^*$:Im | 0.182 | 0.122 | 1.487 | -0.061 | 0.425 | 0.141 |
| $b^*$:Ob | -0.089 | 0.024 | -3.646 | -0.136 | -0.041 | **<0.001**\*\*\* |
| $L^*$:Im:Ob | 0.079 | 0.046 | 1.707 | -0.012 | 0.169 | 0.088 |
| $a^*$:Im:Ob | 0.137 | 0.068 | 2.030 | 0.005 | 0.270 | **0.042**\* |
| $b^*$:Im:Ob | 0.050 | 0.049 | 1.024 | -0.046 | 0.145 | 0.306 |

*P≤0.05

** P≤0.01

***P≤0.001. Im = Image ethnicity, Ob = Observer ethnicity.

**Table 4. Linear mixed effects model estimates of fixed effects, their SE, t-value, lower (2.5%) and upper (97.5%) confidence intervals and P-values for healthiness.**

| Fixed effects | Estimate | SE | t-value | 2.5% CI | 97.5% CI | P-value |
|---|---|---|---|---|---|---|
| (Intercept) | 4.829 | 4.998 | 0.966 | -5.088 | 14.745 | 0.337 |
| L* | -0.015 | 0.065 | -0.231 | -0.144 | 0.114 | 0.818 |
| a* | -0.100 | 0.095 | -1.055 | -0.289 | 0.088 | 0.295 |
| b* | 0.093 | 0.068 | 1.357 | -0.043 | 0.229 | 0.179 |
| Im | -20.374 | 9.995 | -2.039 | -40.205 | -0.544 | **0.045**[*] |
| Ob | 2.141 | 1.725 | 1.241 | -1.242 | 5.523 | 0.215 |
| Im:Ob | -4.515 | 3.434 | -1.315 | -11.248 | 2.218 | 0.189 |
| L*:Im | 0.300 | 0.130 | 2.306 | 0.042 | 0.558 | **0.024**[*] |
| L*:Ob | 0.027 | 0.022 | 1.208 | -0.017 | 0.071 | 0.227 |
| a*:Im | 0.133 | 0.190 | 0.697 | -0.245 | 0.510 | 0.488 |
| a*:Ob | -0.141 | 0.033 | -4.309 | -0.205 | -0.077 | **<0.001**[***] |
| b*:Im | 0.137 | 0.137 | 1.002 | -0.134 | 0.409 | 0.320 |
| b*:Ob | -0.147 | 0.024 | -6.237 | -0.193 | -0.101 | **<0.001**[***] |
| L*:Im:Ob | 0.060 | 0.045 | 1.339 | -0.028 | 0.147 | 0.181 |
| a*:Im:Ob | 0.102 | 0.065 | 1.563 | -0.026 | 0.231 | 0.118 |
| b*:Im:Ob | 0.034 | 0.047 | 0.726 | -0.058 | 0.126 | 0.468 |

[*]P≤0.05

[**] P≤0.01

[***]P≤0.001. Im = Image ethnicity, Ob = Observer ethnicity.

image ethnicity, interactions between image ethnicity and lightness (p = 0.017) and interactions between observer ethnicity and lightness/redness/yellowness are significant (p = 0.001/<0.001/<0.001). The interactions indicate that the effect of lightness on attractiveness is different when faces of different ethnicity are viewed and the effects of lightness/redness/yellowness on attractiveness are different for western European observers and for Chinese observers. The regression lines in S1 Fig in S1 File show all these interactions between colour and ethnicity (observer/image) in the linear mixed-effect models.

Table 4 shows the results of the main effects in the full perceived healthiness model. For perceived healthiness, there was no significant main effect of skin coloration, but a significant effect of image ethnicity. We also found significant interactions between image ethnicity and lightness (p = 0.024) and between observer ethnicity and redness and yellowness, respectively (p<0.001; p<0.001). Similar to facial attractiveness, the effect of lightness on perceived healthiness is different to faces of different origins and the effect of redness/yellowness on healthiness is different for the two groups of observers.

For estimated age, as shown in Table 5, there are significant main effects of redness (a*), image ethnicity, and the interaction between the ethnicity of the image and observer. Significant main effects of interactions between colorations and ethnicity include the interaction between lightness and image ethnicity (p = 0.015), the interaction between redness and image ethnicity (p = 0.036) and the three-way interaction of L*:Im:Ob (p = 0.035). Facial lightness/redness have different effects on the perceived age of European faces and Chinese faces, and the influence of lightness on age perception also depends on the ethnicity of the observer.

To further understand the interactions above and reveal the simple effects of L*, a* and b* within each set of image and observer, parameter estimates for the fixed effects within each subgroup are computed from the three linear mixed-effect models, as shown in Table 6. When Chinese observers rated Chinese faces, an increase in lightness is strongly associated with greater attractiveness (p = 0.002); for western European faces, a decrease in redness predicts

**Table 5. Linear mixed effects model estimates of fixed effects, their SE, t-value, lower (2.5%) and upper (97.5%) confidence intervals and P-values for estimated age.**

| Fixed effects | Estimate | SE | t-value | 2.5% CI | 97.5% CI | P-value |
|---|---|---|---|---|---|---|
| (Intercept) | 23.676 | 14.317 | 1.654 | -4.731 | 52.082 | 0.102 |
| L* | -0.112 | 0.186 | -0.603 | -0.482 | 0.257 | 0.548 |
| a* | 0.550 | 0.273 | 2.017 | 0.009 | 1.091 | **0.047**[*] |
| b* | 0.237 | 0.196 | 1.210 | -0.152 | 0.626 | 0.230 |
| Im | 70.588 | 28.617 | 2.467 | 13.805 | 127.37 | **0.016**[*] |
| Ob | 4.551 | 8.080 | 0.563 | -11.290 | 20.392 | 0.573 |
| Im:Ob | 31.712 | 16.046 | 1.976 | 0.254 | 63.170 | **0.048**[*] |
| L*:Im | -0.926 | 0.372 | -2.487 | -1.665 | -0.187 | **0.015**[*] |
| L*:Ob | -0.072 | 0.104 | -0.693 | -0.277 | 0.132 | 0.488 |
| a*:Im | -1.165 | 0.545 | -2.136 | -2.247 | -0.083 | **0.036**[*] |
| a*:Ob | 0.294 | 0.153 | 1.922 | -0.006 | 0.593 | 0.055 |
| b*:Im | -0.602 | 0.392 | -1.536 | -1.380 | 0.176 | 0.129 |
| b*:Ob | -0.142 | 0.110 | -1.294 | -0.358 | 0.073 | 0.196 |
| L*:Im:Ob | -0.440 | 0.209 | -2.106 | -0.849 | -0.030 | **0.035**[*] |
| a*:Im:Ob | -0.133 | 0.306 | -0.434 | -0.732 | 0.467 | 0.665 |
| b*:Im:Ob | -0.313 | 0.220 | -1.425 | -0.744 | 0.118 | 0.154 |

[*]$P \leq 0.05$

[**] $P \leq 0.01$

[***]$P \leq 0.001$. Im = Image ethnicity, Ob = Observer ethnicity.

greater attractiveness (p = 0.018). Observers of European descent associate an increase in yellowness with higher attractiveness, but only when viewing European facial images (p = 0.010). Lighter skin is associated with greater healthiness but only when Chinese observers rate Chinese images (p = 0.037). Western European observers associate an increase in yellowness with healthiness (p = 0.021) when viewing facial images of Chinese. This association is driven by the western European images: an increase in redness is associated with an older perceived age for western European images when viewed by western European (p = 0.033) or Chinese Observers (p = 0.004). Lightness is a strong predictor for perceived youthfulness when Chinese observers rate Chinese faces (p = 0.002). S2 Figs 2–4 in S1 File show the associations between perceptual ratings and skin color for western European observers and Chinese observers. The regression lines are drawn for the significant fixed effects in the linear mixed-effect model.

## Correlations between the perceptual attributes

Ratings of attractiveness and healthiness are highly correlated across both image and observer ethnicities (Table 7, also see S3 Fig 5 in S1 File) but are negatively correlated with estimated age. The latter negative correlations are highly significant for Chinese observers. The strongest negative correlations are observed when Chinese observers rate Chinese image, consistent with interactions between ethnicity and skin coloration cues.

## Discussion

80 color-calibrated images of real western European and Chinese human faces were used to study how facial coloration was utilized by Chinese and western European observers when rating the healthiness, attractiveness, and youthfulness of facial images of their own or the other ethnicity. The lightness and color variations in these images were representative of the color variations in the respective populations [28]. Our main finding is that skin coloration is not a

**Table 6. Parameter estimates of the simple effects in the linear mixed-effect models.**

| Fixed effects | WE observers | CH observers |
|---|---|---|
| **DV = attractiveness** | | |
| **WE images** | | |
| Model | | |
| $L^*$ | β = -0.140, P = 0.149 | β = -0.102, P = 0.291 |
| $a^*$ | β = -0.090, P = 0.507 | **β = -0.324, P = 0.018**[*] |
| $b^*$ | β = 0.072, P = 0.425 | β = -0.041, P = 0.647 |
| **CH images** | | |
| Model | | |
| $L^*$ | β = 0.105, P = 0.133 | **β = 0.221, P = 0.002**[**] |
| $a^*$ | β = 0.029, P = 0.790 | β = -0.068, P = 0.538 |
| $b^*$ | **β = 0.229, P = 0.010**[**] | β = 0.166, P = 0.059 |
| **DV = healthiness** | | |
| **WE images** | | |
| Model | | |
| $L^*$ | β = -0.164, P = 0.131 | β = -0.166, P = 0.124 |
| $a^*$ | β = -0.071, P = 0.638 | β = -0.263, P = 0.083 |
| $b^*$ | β = 0.106, P = 0.292 | β = -0.058, P = 0.567 |
| **CH images** | | |
| Model | | |
| $L^*$ | β = 0.106, P = 0.170 | **β = 0.163, P = 0.037**[*] |
| $a^*$ | β = 0.011, P = 0.930 | β = -0.079, P = 0.519 |
| $b^*$ | **β = 0.226, P = 0.021**[*] | β = 0.097, P = 0.318 |
| **DV = age** | | |
| **WE images** | | |
| Model | | |
| $L^*$ | β = 0.277, P = 0.381 | β = 0.424, P = 0.180 |
| $a^*$ | **β = 0.952, P = 0.03**[*] | **β = 1.312, P = 0.004**[**] |
| $b^*$ | β = 0.531, P = 0.074 | β = 0.545, P = 0.066 |
| **CH images** | | |
| Model | | |
| $L^*$ | β = -0.429, P = 0.060 | **β = -0.721, P = 0.002**[**] |
| $a^*$ | β = -0.146, P = 0.684 | β = 0.081, P = 0.821 |
| $b^*$ | β = 0.085, P = 0.763 | β = -0.213, P = 0.451 |

[*]$P \leq 0.05$

[**] $P \leq 0.01$

[***]$P \leq 0.001$. DV = dependent variable.

universal predictor for perceived attractiveness, health, or age, but is utilized differently by Chinese and western European observers when rating own- or other-ethnicity facial images.

## Perceived attractiveness and healthiness

Ratings of attractiveness and healthiness are driven by the same facial cues (Tables 3,4,6), which is also supported by the high correlations between attractiveness and healthiness ratings (Table 7), consistent with Han et al. [39]. However, these cues are utilized differently by Chinese and western European viewers. Chinese observers use skin lightness as a robust cue for judging both attractiveness and healthiness, but only when judging Chinese faces (Table 6). In

**Table 7. The Pearson Correlation Coefficients of age, healthiness, and attractiveness scores for the western European (WE) and Chinese (CH) observers.**

|  | WE images | CH images | Overall images |
|---|---|---|---|
| **WE observers** |  |  |  |
| Attractiveness-Healthiness | 0.912*** | 0.946*** | 0.929*** |
| Attractiveness-Age | -0.343 | -0.354 | -0.351* |
| Healthiness-Age | -0.293 | -0.295 | -0.298 |
| **CH observers** |  |  |  |
| Attractiveness-Healthiness | 0.881*** | 0.927*** | 0.893*** |
| Attractiveness-Age | -0.632*** | -0.828*** | -0.730*** |
| Healthiness-Age | -0.651*** | -0.818*** | -0.726*** |

*$P \leq 0.05/18$

** $P \leq 0.01/18$

***$P \leq 0.001/18$. N = 40, 40, 80 for WE, CH and overall images, respectively. All p-values were Bonferroni-corrected.

contrast to our study, Han et al. [39] found an effect of lightness on attractiveness/perceived health for both, Chinese and Western faces, whereas in our study we find significant associations only when Chinese faces were rated. A possible explanation for this discrepancy is that Chinese skin is characterized by a lower average $L^*$ value, and crucially, by a smaller variation in lightness compared to the skin of western Europeans [40]. We speculate that it is the smaller variability in $L^*$ for Chinese faces, which leads to a more informative and more reliable lightness cue.

For Chinese viewers, redness ($a^*$) is negatively associated with both attractiveness and healthiness but reaches significance ($p < 0.05$) only for attractiveness when viewing European faces (Table 6). In contrast, western European viewers associate an increase in yellowness ($b^*$) with increased attractiveness and healthiness, but only when viewing images of Chinese faces (Table 6). The association between an increase in skin yellowness ($b^*$) and perceived attractiveness and healthiness is likely to reflect the western European preference for 'tanned' skin which is characterized by a concurrent increase in $b^*$ and decrease in $L^*$ [41]. Skin yellowness as a significant predictor for perceived health is consistent with previous studies (e.g. [42]). Whether skin yellowness is associated with physical health is controversial; some studies found correlations between physical fitness and skin yellowness [43], whereas others found associations between skin yellowness and perceived health, but not with physical health [10]. Experiments linking high-carotenoid diet to objective skin color changes have reported consistent associations between carotenoids intake and skin yellowness ($b^*$), however, the results for the carotenoids-redness ($a^*$) linkage are inconclusive. Fruit and vegetable (FV) intake has been associated with a significant increase in skin yellowness (about 2 $b^*$ units, much smaller in $a^*$ values) and perceived attractiveness, consistent with the putative carotenoid-linked health-signaling system for mate choice [12, 44]. Pezdirc et al [45] reported an increase of 0.6 $b^*$ but no measurable increase in redness ($a^*$), which was confirmed by Appleton et al. [8] in a randomized controlled trial where they documented a small but significant effect of FV intake on skin yellowness (about 1–2 $b^*$ units), but no change in skin redness and no effect on perceived health. On the contrary, Tan et al. reported significant increments in both skin yellowness and redness ($p < 0.001$) after 4-week FV intervention [46]. Highly elevated skin yellowness levels are likely to reflect underlying health issues such as jaundice [47]. Therefore, one would predict that only those small changes in yellowness that reflect objectively measurable health benefits, to be positively associated with *perceived* health. This non-linear relationship between physical skin coloration changes and perceptual preferences supports the argument that associations

between physical skin changes and perceptual attributes such as healthiness and attractiveness need to be investigated within a physically plausible, evolutionary relevant, parameter range.

## Perceived age

The strongest negative associations are found between skin lightness and estimated age, but only when Chinese observer rate own-ethnicity faces (Table 6). This robust relationship between skin lightness and perceived youthfulness in the Chinese culture may reflect the physical lightness changes as function of age: a decrease of one $L^*$ unit is equivalent to a 10-year increase in age in female Chinese skin [48]. Western European observers, on the other hand, show no robust associations between skin lightness and any of the measured attributes. Skin lightness as a cue to attractiveness and youthfulness is therefore deployed differently in these two cultures. For western European images, facial redness ($a^*$) is positively associated with estimated age, irrespective of the ethnicity of the viewer (Table 6). The role of facial redness is discussed in more detail below.

## The role of other facial cues

Two facial coloration cues are used robustly, but differentially between ethnicities: skin lightness ($L^*$) within the Chinese sample as an indicator for attractiveness, health and youthfulness, whereas skin redness ($a^*$) is negatively associated with youthfulness in western European facial images (Table 6). Since we used non-manipulated images of real faces, skin coloration co-varied necessarily with other facial shape features. The high observer consistency both within and across ethnicities (Table 1) suggest that observers may rely on additional facial cues in their judgments, consistent with Fink et al.'s study [49] in which ratings for perceived age, health, and attractiveness were obtained for full facial images and for isolated cropped cheek patches. Associations between these two sets of ratings were modest: $R^2$ were 0.31 (perceived age), 0.14 (health) and 0.06 (attractiveness), which demonstrates that participants must make use of other facial features (not contained in the isolating skin patch) when rating the skin patches/ faces along these attributes. In the Fink study, the most robust association were reported for perceived age, where the isolated skin image explains about 31% of the variation in the full-face image ratings, consistent with the melanin distribution being a strong indicator of youthfulness [50]. Our current study was designed to estimate the strength of association between mean skin coloration ($L^*$, $a^*$, $b^*$) and the perceptual ratings of perceived health, attractiveness, and perceived age; it does not allow us to estimate the contribution of skin coloration relative to the contribution of other cues, such as skin texture, localized coloration changes or other anatomical facial features. A useful follow-up study would be to include in the linear fixed-effect model spatial location and skin texture as independent variables, which would allow us to evaluate the relative contributions of mean skin coloration vs $2^{nd}$-order skin characteristics.

## No evidence for a positive association between skin redness and perceived attractiveness, health, and youthfulness

We find no evidence that facial skin redness is positively associated with perceived attractiveness, healthiness or youthfulness (see Table 6), in contrast to previous reports with western European (Stephen et al., 2009a; Stephen et al., 2009b; Pazda et al., 2016; Stephen et al., 2012) [2, 3, 13, 18] and Chinese observers (Han et al., 2018; Tan et al., 2019) [20, 21]. Our results are, however, consistent with recent studies using a large image data base of real female faces that did not find any association between facial redness and objective health measures, neither any positive association between redness and attractiveness [9]. Other studies employing non-manipulated real facial images, found a weak positive association between skin yellowness and

facial attractiveness, but skin redness as a mediator showed a small but negative association with facial attractiveness [8]. Facial redness, caused by increased blood oxygenation, has been postulated to be an important signal for mating choice, by serving as an indicator for fertility [51–53], but recent evidence suggests a more complex relationship between female facial redness and its role in signaling fertility. While perceived attractiveness and healthiness has been shown to be higher in the follicular phase, the changes in redness (a*) were either not measurable [54], below perceptual threshold [55] or not cycle-specific [56], which suggests that other facial cues, such as facial shape and skin homogeneity, may play a more prominent role.

Studies that have reported a strong positive association between facial redness and attractiveness or healthiness, have involved color-manipulated facial images [2, 3, 12, 13]. More recent studies, including the current one, using non-manipulated facial images, have failed to show these strong associations [8–11]. This discrepancy could be partly due to methodological differences; including the magnitude of redness changes (in excess of 10 a* units) and color shifts being applied uniformly across the face. Skin color manipulations were often restricted to the CIELAB dimensions, a*, b*, L*, whereas in the natural skin color universe, skin color dimensions are highly correlated [28, 43]. Crucially, changes in skin color reflecting evolutionary relevant or life-style changes (fertility, exercise, diet, physical health) are not restricted to one of these dimensions, but are characterized by co-variations along all three dimensions (e.g. [8, 55]). It is conceivable that our sample of facial images (n = 80) did not contain a sufficiently large color range, in particular redness variations, to reveal associations with the subjective ratings. We believe this is unlikely to explain our results, since the range of redness values covered by our images is about 6 a* units (Fig 2.), whereas, for color-normal observers, two a* units are easily discriminable for facial skin patches [4, 57]. A potential limitation of our study is that we used average skin color as a predictor instead of using the color values at specific facial locations, which has been shown to be an important factor [14].

All these methodological differences taken together may explain some of the discrepancies in the suggested role of skin coloration for subjective ratings of health and attractiveness. Broadly speaking, rating experiments using manipulated facial images showed strong associations [2, 3, 12, 19, 20, 58] between skin coloration and subjective ratings of health and attractiveness. While color manipulation of the facial images has the advantage of dissociating the role of skin color cues from facial shape cues, they may not reflect evolutionary significant color changes (as discussed above), and recent studies employing natural, non-manipulated images [8–11, 43] or plausible physics-based skin image manipulations [59, 60] reported much weaker associations between skin color and healthiness and attractiveness, suggesting that the former studies may have led to an overestimation of the role of skin color for health and attractiveness, particularly for associations with facial redness. Jones [11] concluded that perceived health and attractiveness relies almost exclusively on facial shape features with mean facial color playing a minor role. To fully characterize the differential contributions of facial shape, average skin color and skin homogeneity requires all features to be manipulated independently, but within an evolutionary meaningful parameter space, ideally using a realistic skin model based on physical parameters including chromophore distribution and magnitude [59], blood oxygenation, skin moisture, translucency, sub-surface skin scatter, all of which affect skin appearance and therefore potentially provide potential cues to healthiness, attractiveness, and youthfulness.

In summary, the most robust positive associations were found between facial skin lightness (L*) and attractiveness, healthiness, and youthfulness, but only when Chinese observers judge facial images of their own ethnicity. These associations between ratings and skin lightness are grounded in known physical changes: a decrease of one L* unit is equivalent to a 10-year increase in age in female Chinese skin [48]. In contrast to previous studies, we find no evidence

that facial redness is positively associated with perceived attractiveness or health. A possible explanation for this discrepancy with previous results is that we used naturally occurring skin coloration variations, instead of manipulating the facial images. We speculate that previous experiments may have overestimated these associations by using skin color manipulations well beyond the gamut found in non-manipulated images as well as changing the coloration of the entire face instead of specific areas.

The effect of ethnicity has been investigated in our experiments using observers of both genders, but we appreciate that observer gender is a possible additional variable. While our sample is too small to conduct a meaningful gender-based analysis, previous studies found a strong effect of facial redness that impacts on perceived health and attractiveness for both male and female skin by skin color manipulations [61, 62]. Whether there is a perceptual difference for facial colour appearance between gender and, if there is, how large the effect is in the realistic skin model compared to the cultural difference, requires further work.

We conclude that observers of both ethnicities make use of skin color and lightness to rate attractiveness, healthiness, and perceived age, but the utilization of these cues is more subtle than previously thought. Crucially, skin coloration cues are not universal and are utilized differently within the Chinese and western European ethnic groups, reflecting different aesthetic preferences in eastern and western cultures. Such ethnic differences in objective aesthetic criteria should be considered in many applications of preferred skin colour reproduction including aesthetic surgery [63].

## Supporting information

**S1 File.**
(RAR)

## Author Contributions

**Conceptualization:** Yan Lu, Kaida Xiao, Michael Pointer, Changjun Li, Sophie Wuerger.

**Data curation:** Yan Lu, Jie Yang, Kaida Xiao.

**Formal analysis:** Yan Lu, Jie Yang, Changjun Li, Sophie Wuerger.

**Funding acquisition:** Sophie Wuerger.

**Investigation:** Yan Lu, Michael Pointer, Changjun Li.

**Methodology:** Yan Lu, Jie Yang, Kaida Xiao, Michael Pointer, Sophie Wuerger.

**Project administration:** Michael Pointer.

**Supervision:** Kaida Xiao, Michael Pointer, Sophie Wuerger.

**Validation:** Yan Lu.

**Visualization:** Yan Lu, Changjun Li.

**Writing – original draft:** Yan Lu, Kaida Xiao.

**Writing – review & editing:** Yan Lu, Kaida Xiao, Michael Pointer, Changjun Li, Sophie Wuerger.

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
