## [Decision Letter · Decision Letter 0]

24 Jun 2021

PONE-D-21-13133

Skin coloration is a culturally-specific cue for attractiveness, healthiness and youthfulness in observers of Chinese and Western European descent.

PLOS ONE

Dear Professor Wuerger,

Dear Professor Wuerger,

Thank you for submitting your work to PLOS ONE. I have now had chance to secure the opinion of two reviewers with significant expertise in the area of both mixed models, and the role of facial appearance and colouration in social perception.

As you will see, the reviews are quite favourable, and centre mostly around including more detail in the introduction and method, as well as more clearly illustrating the results from the mixed models. Reviewer 1 highlighted that some of the results were quite confusing (as can often be the case with mixed models) and looking at the raw data may clear things up, but was unable to access it. I would strongly encourage the authors to deposit their raw data somewhere that can be accessed, even if only for review purposes.

I hope the authors find these comments useful. I am in the unusual position of acting as a handling editor after reviewing an earlier version of this manuscript, and I find that while I agree with both reviewers comments, I find it significantly improved and commend the authors on the hard work they have put in already.

I look forward to receiving a revised manuscript in due course.

We look forward to receiving your revised manuscript.

Kind regards,

Alex Jones

Academic Editor

PLOS ONE

Journal Requirements:

2. Please note that in order to use the direct billing option the corresponding author must be affiliated with the chosen institute. Please either amend your manuscript to change the affiliation or corresponding author, or email us at plosone@plos.org with a request to remove this option.

3. We note that Figure 1 includes an image of a participant in the study. 

Reviewers' comments:

Reviewer's Responses to Questions

**Comments to the Author**

1. Is the manuscript technically sound, and do the data support the conclusions?

Reviewer #1: Partly

Reviewer #2: Yes

2. Has the statistical analysis been performed appropriately and rigorously? 

Reviewer #1: I Don't Know

Reviewer #2: I Don't Know

3. Have the authors made all data underlying the findings in their manuscript fully available?

Reviewer #1: No

Reviewer #2: Yes

4. Is the manuscript presented in an intelligible fashion and written in standard English?

Reviewer #1: Yes

Reviewer #2: Yes

5. Review Comments to the Author

Reviewer #1: The current manuscript investigates cultural differences in the preference for and social perception of facial coloration in a sample of Chinese and Western European participants. I think there is great value in complementing existing findings on experimentally manipulated images with work on unmanipulated, natural images. However, there are several points I would like to see the authors address before I would recommend this manuscript for publication.

General comments

- I would ask the authors to make their data and analysis script(s) available for the reviewing process; I could not find these at the provided link. I have several questions regarding the analyses (see below), some of which I might have been able to answer myself with access to the data

- Han et al. (2018) found effects of both stimulus and observer sex. I assume the authors did not test for such effects due to sample constraints, but this should be acknowledged somewhere

Materials and Methods

Stimuli:

- How were images selected from the LLSD? Demographics of stimulus set should be reported (could age of stimulus faces have had an effect on results?)

- What software was used to mask non-face cues, how were images scaled/centered (i.e. algorithmically or manually), and how were CIELAB coordinates calculated?

Observers:

- How were observers recruited?

- It is stated Chinese observers had spent about 1-3 years in the UK – mean and SD would be more informative here

Procedure:

- Was the order of sessions randomized?

Statistical analysis:

- It is stated that “all interactions between the predictors were included”—this could be a bit clearer. Was a full factorial design specified? Only two-way interactions are reported in Table 2, while Table 3 and the text suggests three-way interactions were analysed (see also comments below)

Results

- It is not clear exactly which models were tested and how they were specified. Table 2 reports main effects and two-way interactions, while Table 3 is split by observer ethnicity. This only make sense if the respective three-way interactions were included in the main model and found to be significant, in which case the corresponding stats need to be reported. Moreover, Table 2 only presents F- and p-values, while Table 3 reports significant estimates. All of this makes it hard to evaluate results and the subsequent discussion of effects. Findings would also be easier to parse if they were presented separately for the three DVs. This extends to Tables 2 and 3: estimates (including 95% CIs) and corresponding stats should be presented together, and tables instead split by DV

- It would be helpful to include graphs of significant interactions of interest (either in the main manuscript or the supplemental material)

- When describing results in-text, effect sizes should be reported in addition to p-values

- Results should maybe start with info on inter-rater agreement (since low inter-rater agreement would render any further analyses meaningless). In that section it is also stated observers showed higher agreement within their own ethnicity— as the numbers are only marginally different, I am not sure this warrants extra mention?

- Presenting correlations between perceptual ratings separately for Chinese and Western observers and stimuli suggests that associations between these sub-groups differed—was this tested? It looks like age was more strongly associated with attractiveness and health in Chinese compared to Western participants, but I am not sure differences between Chinese and Western images would actually reach statistical significance?

Discussion

- Like the results, I feel the discussion would benefit from some restructuring. Again, I found it a bit hard to understand what exactly was found and how these results sit within the existing literature

- Attractiveness and b*. The positive effect of b* on attractiveness that was found for Chinese faces as judged by Western observers is interpreted in the context of a Western preference for tanned skin. This begs the question why no such preference was found for Western faces? In addition, there is some evidence that tan/melanin pigmentation is less preferred compared to carotenoid pigmentation. While there likely is some perceptual overlap between the two, the tan explanation does not appear entirely satisfactory. The authors also appear to doubt that high-carotenoid intake is linked to changes in skin pigmentation (line 252), but cite several studies that show such a link? Lastly, b* does not necessarily have to be linked to actual health in order to affect perceived health (or attractiveness); other mechanisms have been suggested (e.g., Han et al., 2018). I would be interested to hear the authors’ thoughts on this.

- Effects of L*. In line with the current study, Han et al. (2018) found a stronger association of L* and attractiveness/perceived health for Chinese compared to Western observers. This also appears consistent with the current finding that L* was linked to perceived age in Chinese observers (especially since ratings of age seem to be quite strongly correlated with ratings of attractiveness and perceived health in Chinese observers). However, Han et al. reported an effect for both Chinese and Western faces, whereas the current study only appears to have found such an effect for Chinese stimulus images; I wonder whether this might have to do with different base levels of L* in unmanipulated images of Chinese and Western faces? Again, I would be interested to hear the authors’ thoughts on this.

- While I agree with the authors that colour cues are perceptually integrated with other facial cues (such as shape), I felt in its current form the section on “the role of other facial cues” was a bit unclear: what are the implications for the current study/findings?

- Effects of a*. The discussion in this section switches in terminology to referring to “youthfulness” instead of age, which I thought was slightly confusing. It is reported that a* negatively affected attractiveness (lines 302 and 358); it should be clarified that this effect was only found for Chinese observers rating Western images. The authors provide some ideas as to why a* was not positively linked to attractiveness and health as in (some) previous studies; I was wondering whether these null- and negative findings on Western images in particular might also be linked to the association of a* and perceived age in the Western image set?

Minor comments:

- Table 2 is labeled main effects but also presents interaction effects

- Table 3 should specify whether the reported estimates are standardized or not

- Tense sometimes switches between past and present in results

- While most references are indexed by numbers, a few are referred to in the FirstAuthor et al. format

- Supplemental info is labeled a bit confusingly; Figs. 1-3 are labeled as S1, and then there is an unnumbered Fig. that is labeled as S2. In addition, it would be good to add a bit more info to figure captions

Reviewer #2: This is an interesting cross-cultures study set to examine the associations between skin colouration and subjective rating of attractiveness, apparent health and perceived age across two independent samples. The authors highlighted the methodological uniqueness (difference) of the current study, as compared to those examined the similar variables. The description of the method, as a whole, is sound, appropriate and with a decent amount of details.

A few minor suggestions and recommendations listed below, for the authors’ consideration:

1. The literature can be further expanded, both in breadth and also depth, to better established the scene – and justify the proposed hypothesis. The authors should consider discussing the potential mechanism involved, in brief. Also, worth discussing the reasons for potential cultural difference – apart from reporting the mixed results.

2. It would be better if the authors can introduce the term LAB-C, in brief, before discussing the individual component using only the label/acronym.

3. The presentation (display) of the stimulus should be better described for clarity. What software did the researcher use? And how was the actual protocol like? This info seems to be immersed with the image preparation part.

4. Please refer to line 132 – Were the participants being asked to attend only to the skin colour part of the image and not the overall face? If so, justification may be needed for this unconventional approach considering that there is a substantial amount of evidence showing that face is being processed holistically. If that is not the case, the writer may need to rephrase the specific statement.

5. It would be great if the writer can provide some simple justification about the sample size used – to ensure that it has enough power.

6. Refer to line 112 - It may be better if the author can explicitly report the mean values of LAB for the Asian and Caucasian faces used – instead of a generic description about the difference between the two sets of stimuli. Even the actual mean difference will be better.

7. Please refer to line 206-210 – The Result can be better phrased for clarity. Phrases such as “highly significant” or “highest negative correlation” may not be the conventional way of reporting the result – or at least not specific enough and the point of comparison is not clear. Please consider rephrasing this section for clarity.

8. Please refer to line 213 – I am not certain what the number 18 after the p values stand for.

9. Please refer to line 218-220 – A stat may be needed to support the inference made about the higher agreement of own ethnic faces rating than of the other ethnic’s.

10. Line 221-223, please explain the number (22)(44) in the bracket?

11. Line 228, maybe reconsider if the term “deployed” is best to be used here. This is merely a suggestion.

12. Table 3 is incomplete and hard to be seen clearly. I would assume the authors will also need to report those stats with a not-significant result, at least in the table.

13. Line 240-242, not sure if the conclusion stated here is accurate – I would refrain from implying that one (aspect) is higher than another if the result is indicating otherwise.

14. Line 252-253 – the author reported that - “Experiments linking high-carotenoid diet to objective skin color changes have produced inconclusive results.” However, (all) the evidence listed subsequently in the adjacent sentences seems to be indicating that the supplementation consistently induces an increment in skin yellowness. The inconclusive part, I assume should be more on redness (and the subjective rating of health)?

15. Line 260-263 – please consider rephrase this section, for better clarity.

16. I would avoid using the comparative terms (highest, higher etc) especially when there is no obvious reason for doing so (hypothesis wise, theoretically wise). Instead of saying “highest correlation”, a better way to go may be to merely indicate the valence (positive) and the strength (weak, medium, strong) – see, for example, line 265

17. Discussion can be expanded and better organized – This section seems to be mostly descriptive (the who and what – e.g., who found what). I would expect some explanation about the potential mechanism here (the why and how) – at least in brief. For example, why is luminance being associated with higher perceived health (and/or physical health)? Why redness is being related to the increment in perceived age?

18. Please refer to line 300 – the authors wrote ‘also failed to find any association between facial redness and objective health measures”. This may be implying that the authors were also measuring the objective health measures – which I assume is not the case.

19. Line 301, would it be better to explicitly cite the last name here – for clarity and ease of reading. This is merely a suggestion.

20. Line 301-302 may not be entirely accurate, as redness is not, as a whole, correlated with attractiveness.

21. Please check the in-text citation on Line 331.

22. Line 355-356, this argument needs to be rephrased – as it is unclear what do you mean by “manipulated facial images showed strong association”. You have mentioned the stimuli used here (facial images) but not the IVs and DVs.

23. Please check your reference list – a few items do not seem to be with complete info (e.g, item 2, 8, 16, 21, 22, 23, 28, 32, 44).

6. PLOS authors have the option to publish the peer review history of their article (what does this mean?). If published, this will include your full peer review and any attached files.

Reviewer #1: No

Reviewer #2: **Yes: **Tan Kok Wei

---

## [Author Response · Author response to Decision Letter 0]

5 Aug 2021

We appreciate that both reviewers and editor Alex Jones have given us a lot insightful comments and valuable feedback. We have revised our manuscript according to the comments and suggestions (please find the manuscripts with and without track changes), and we have also responded to each point raised by reviewers (please see 'Response to Reviewers'). Again, thanks both reviewers and editor who really made us think more and helped us improve our paper. We hope the revised paper is improved and any new comments and suggestions are always welcome.

---

## [Decision Letter · Decision Letter 1]

1 Oct 2021

PONE-D-21-13133R1

Skin coloration is a culturally-specific cue for attractiveness, healthiness and youthfulness in observers of Chinese and Western European descent.

PLOS ONE

Dear Dr. Wuerger,

Thank you for submitting your manuscript to PLOS ONE. After careful consideration, we feel that it has merit but does not fully meet PLOS ONE’s publication criteria as it currently stands. Therefore, we invite you to submit a revised version of the manuscript that addresses the points raised during the review process.

We look forward to receiving your revised manuscript.

Kind regards,

Alex Jones

Academic Editor

PLOS ONE

Journal Requirements:

Additional Editor Comments (if provided):

Dear Professor Wuerger,

Apologies for the delay in returning reviews to you; it is a busy time of year.

Both the original reviewers have now returned their comments to me on the revised version of the manuscript. As you can see, both are generally pleased and find the work much improved, with only a few outstanding comments. I am recommending a minor revision here to address these final points, and will not be sending the manuscript back out for further review once these have been completed.

Reviewers' comments:

Reviewer's Responses to Questions

**Comments to the Author**

1. If the authors have adequately addressed your comments raised in a previous round of review and you feel that this manuscript is now acceptable for publication, you may indicate that here to bypass the “Comments to the Author” section, enter your conflict of interest statement in the “Confidential to Editor” section, and submit your "Accept" recommendation.

Reviewer #1: (No Response)

Reviewer #2: (No Response)

2. Is the manuscript technically sound, and do the data support the conclusions?

Reviewer #1: Partly

Reviewer #2: Yes

3. Has the statistical analysis been performed appropriately and rigorously? 

Reviewer #1: No

Reviewer #2: Yes

4. Have the authors made all data underlying the findings in their manuscript fully available?

Reviewer #1: Yes

Reviewer #2: Yes

5. Is the manuscript presented in an intelligible fashion and written in standard English?

Reviewer #1: Yes

Reviewer #2: Yes

6. Review Comments to the Author

Reviewer #1: I appreciate the changes the authors’ have made in response to both reviewers. There are a few more points I’d like to see them address:

• The authors have now shared their data file, which is great! I wonder whether they also plan to share their R analysis script? This would make it much easier to check analyses and suggest any potential improvements. I also note that I still could not find any data specific to the manuscript under the link they provide

• The authors have now added some notes on sex of the images in the experimental procedure. I would still ask them to explicitly note this as a limitation of the current study; I’m not convinced that masking non-face cues actually also effectively masks any cues to sex/makes sex irrelevant in perception

• The authors have explained in their response that all faces were between the ages of 20 and 40. Just to clarify, did they use all faces that fell within this age bracket, or were there any other criteria for inclusion/exclusion (so if someone wanted to replicate this study using the same stimulus set they’d have all the info they needed)? Would it maybe be possible to report mean age of the used faces?

• It still wasn’t entirely clear to me whether masking and scaling were done manually or using some sort of algorithm. Either is fine, but in the interest of reproducibility, this could be explained in more detail

• Analyses/reporting of results. The paragraph on Statistical Analysis (lines 167 ff) should include info on effect-coding (how were image- and observer-ethnicity coded). Table 3 should also include estimates and 95% CIs (I can provide code for that if the authors want). Minor point, but p-values <.001 should be reported as such (instead of p=.000), and leading 0s should be omitted.

• I recommend breaking up reporting of the three DVs into separate subsections to make them easier to digest. Similar to my comment in the previous round, Table 2 should include corresponding estimates, and analyses should not be split by image and observer ethnicity (as for Table 4, and interpretation of effects). For example, re the significant two-way interactions of L*/b* x observer ethnicity on attractiveness: There is a positive effect of l* for Chinese but not Western observers; there is a positive effect of b* that’s even stronger in Western faces; but the model does not indicate these effects differ significantly for Chinese and Western observers. The three-way interactions are easily enough explained by visualizing them—I’ve attached an example of the sort of graph I have in mind (left panel Western observers, right panel Chinese observers)

• The authors have now included F-values in Table 3, but not really addressed my comment—I meant that, e.g., on line 209, “interaction between image ethnicity and lightness was significant (p=0.017)” should instead be “(estimate=0.28, 95%CI [0.06, 0.51], t78.74=2.45, p=.017)”

Reviewer #2: The manuscript improved significantly after the revision. Specifically, for the Introduction, the rationale has been strengthened. A substantial amount of details have also been added into the Method section – which enhances both the clarity and replicability. The result section is also clear and is supported with appropriate figures. Discussion is also more comprehensive now.

A few suggestions for minor change were as followed,

1. Argument on Lines 73-76 should be supported with evidence.

2. For attractiveness, I reckon there is a significant effect of image ethnicity? Please check. See Lines 207-210 and Table 3.

3. For estimated age, is there is a significant interaction between the ethnicity of the image and observer? If so, is there any reason why it was not being included in the writing?

4. Western European observers associate an increase in yellowness with healthiness – only when viewing facial images of Chinese. Can you please check again your writing in the Result: Lines 232-233 (or maybe Lines 232-235, as a whole).

5. In Discussion, Line 276-277, the claim made here should only be relevant for the viewing of the European face. Please check.

6. For the Discussion, the authors have not properly fixed the carotenoids-yellowness-redness linkage. The authors start their claim by saying that the observed change in skin redness with the consumption of carotenoids was not consistent (in the lit). They then reported only those studies that observed NO significant change in skin redness (with the consumption of carotenoids). Hence the evidence presented does not seems to be matching the core argument. Perhaps this paper can be used to illustrate the “inconsistent” part - https://pubmed.ncbi.nlm.nih.gov/26186449/

7. Also, since carotenoids colouration is related to both skin yellowness and skin redness – I would suggest the author start the sub-section by saying that – “even though the association between carotenoids consumption and skin yellowness has been consistently reported in previous studies, but such consistency cannot be applied to the linkage between carotenoids intake and skin redness (with their phrasing of course). It seems awkward to include a claim merely highlighting skin redness in between two chunks of discussion about skin yellowness. See Lines 277-301 but specifically also the claim made on Lines 287-288.

Thank you.

7. PLOS authors have the option to publish the peer review history of their article (what does this mean?). If published, this will include your full peer review and any attached files.

Reviewer #1: No

Reviewer #2: **Yes: **Tan Kok Wei

---

## [Author Response · Author response to Decision Letter 1]

14 Oct 2021

We really appreaciate the editor, Alex Jones, for allowing the revision of our manuscript, with an opportunity to address both reviewers' comments. The comments from both reviewers are very detailed and really valuable for helping us to improve our manuscript. We have read all the comments and have revised our manuscript accordingly. We are uploading our point-by-point response to the comments, an updated manuscript with all the changes marked in red, and a clean updated manuscript without marks. Please review the changes and feel free to get back to us with any further questions/suggestions.

---

## [Editor Report · Decision Letter 2]

18 Oct 2021

Skin coloration is a culturally-specific cue for attractiveness, healthiness, and youthfulness in observers of Chinese and Western European descent.

PONE-D-21-13133R2

Dear Dr. Wuerger,

We’re pleased to inform you that your manuscript has been judged scientifically suitable for publication and will be formally accepted for publication once it meets all outstanding technical requirements.

Kind regards,

Alex Jones

Academic Editor

PLOS ONE
---

## [Editor Report · Acceptance letter]

20 Oct 2021

PONE-D-21-13133R2 

Skin coloration is a culturally-specific cue for attractiveness, healthiness, and youthfulness in observers of Chinese and western European descent 

Dear Dr. Wuerger:

I'm pleased to inform you that your manuscript has been deemed suitable for publication in PLOS ONE. Congratulations! Your manuscript is now with our production department. 

Kind regards, 

on behalf of

Dr. Alex Jones 

Academic Editor

PLOS ONE